# Immunotherapy in Glioblastoma: Current Approaches and Future Perspectives

**DOI:** 10.3390/ijms23137046

**Published:** 2022-06-24

**Authors:** Ugur Sener, Michael W. Ruff, Jian L. Campian

**Affiliations:** 1Department of Neurology, Mayo Clinic, Rochester, MN 55905, USA; ruff.michael@mayo.edu; 2Department of Medical Oncology, Mayo Clinic, Rochester, MN 55905, USA; campian.jian@mayo.edu

**Keywords:** glioblastoma, immunotherapy, checkpoint inhibitor, CAR T, vaccine therapy, viral therapy, cytokine-based therapy

## Abstract

Glioblastoma (GBM) is the most common malignant brain tumor. Despite multimodality treatment with surgical resection, radiation therapy, chemotherapy, and tumor treating fields, recurrence is universal, median observed survival is low at 8 months and 5-year overall survival is poor at 7%. Immunotherapy aims to generate a tumor-specific immune response to selectively eliminate tumor cells. In treatment of GBM, immunotherapy approaches including use of checkpoint inhibitors, chimeric antigen receptor (CAR) T-Cell therapy, vaccine-based approaches, viral vector therapies, and cytokine-based treatment has been studied. While there have been no major breakthroughs to date and broad implementation of immunotherapy for GBM remains elusive, multiple studies are underway. In this review, we discuss immunotherapy approaches to GBM with an emphasis on molecularly informed approaches.

## 1. Introduction

Glioblastoma (GBM) is the most common primary brain tumor, representing 14.5% of central nervous system (CNS) tumors and 48.6% of malignant CNS tumors in the United States [1,2]. According to Central Brain Tumor Registry of the United States (CBTRUS), incidence is 3.23 per one hundred thousand with an average of 12,011 cases diagnosed per year between 2013 and 2017 [2]. Treatment of GBM involves surgical resection followed by radiation therapy (RT) with concomitant and adjuvant temozolomide chemotherapy [1,3]. Epigenetic silencing of DNA-repair gene O6-methylguanine–DNA methyltransferase (MGMT) is associated with improved overall survival and increased benefit from temozolomide chemotherapy [4]. Addition of tumor-treating fields to temozolomide has been associated with modest additional survival benefit [5]. Despite multimodality treatment, tumor recurrence is universal with median observed survival low at 8 months and 5-year overall survival (OS) poor at 7% [2]. There is a great need for novel treatment approaches that will improve patient outcomes.

Immunotherapy aims to generate a tumor-specific immune response to selectively eliminate tumor cells. Immunotherapy with checkpoint inhibitors and chimeric antigen receptor (CAR) T-Cell therapy has been effective in treatment of solid organ and hematologic malignancies generating interest in application of these techniques for patients with GBM [6,7,8,9,10]. However, this has been tempered by the notion that the CNS is an immunoprivileged environment and recognition of the immunosuppressive nature of GBM [11,12,13].

Traditionally, the CNS has been viewed as an immunoprivileged environment monitored by microglia, in part due to the presence of the blood–brain barrier and the observation of prolonged survival of engrafted tissue in the brain compared to other locations [11,12,13]. However, emerging data suggests the CNS is immunologically distinct rather than immunoprivileged [11,12]. Indeed, there is a complex interplay between the CNS and the immune system, as illustrated by findings such as identification of lymphatic vessels running parallel to dural venous sinuses and autoimmune basis for conditions such as neuromyelitis optica and multiple sclerosis [14,15].

GBM has been described as an immunologically ‘cold’ tumor with multiple immunosuppressive mechanisms [1,16,17,18,19]. In general, GBM is associated with a low mutational burden, representing few neoantigens to elicit an immune response [1]. GBM secretes paracrine immunosuppressive mediators and causes systemic immunosuppression via sequestration of T cells in bone marrow [17,18]. Few tumor-infiltrating lymphocytes are present resected GBM specimens and the existing T cells are associated with an exhausted phenotype [19]. Successful implementation of immunotherapy in GBM requires overcoming these factors [20,21,22]. Here we review approaches to immunotherapy in GBM by highlighting prior studies with checkpoint inhibitors, CAR T-cell therapy, vaccine-based therapies, viral therapies, and cytokine-based approaches with a focus on utilizing the molecular makeup of tumor cells to guide treatment selection.

## 2. Immune Checkpoint Inhibitors

Checkpoint regulators such as cytotoxic T-lymphocyte-associated protein 4 (CTLA-4) and programmed cell death protein 1 (PD-1) downregulate T cell activation [23]. Antigen presenting cells (APC) present peptide fragments to T cells via major histocompatibility complexes [23,24]. Activation of T cell receptors leads to CTLA-4 expression, promoting self-tolerance and preventing autoimmunity [23,25]. Anti-CTLA-4 agents such as ipilimumab and tremelimumab enhance T cell stimulation by blocking this inhibitory pathway [23,25]. Similarly, PD-1 is an inhibitory receptor expressed on T cells with program death receptor ligand (PD-L1) potentially expressed on tumor cells [23,24,25]. Anti-PD-1/PD-L1 agents such as pembrolizumab, nivolumab, and atezolizumab increase T cell activation by blocking this inhibitory pathway [23,24,25].

Checkpoint inhibitors have been studied in the treatment of newly diagnosed and recurrent GBM. In an open-label, randomized, multicenter, phase III clinical trial, 369 patients with recurrent GBM were randomized in 1:1 fashion to treatment with nivolumab vs. bevacizumab [26]. Median OS (mOS) was 9.8 months in the nivolumab group compared to 10 months in the bevacizumab group [26]. In an exploratory post hoc subgroup analysis, patients with MGMT methylated tumors with no baseline corticosteroid had 17.0 month mOS compared to 10.1 month mOS observed for patients with similar tumors treated with bevacizumab, suggesting a subset of patients may benefit from checkpoint inhibitor monotherapy [26].

In a phase III, randomized, multi-center clinical trial, RT with nivolumab was compared to RT with temozolomide for patients with newly-diagnosed, MGMT-unmethylated GBM [27]. The study did not meet its primary endpoint of OS [27]. In a separate phase III, randomized clinical trial, RT with temozolomide and nivolumab was compared to RT with temozolomide for patients with newly diagnosed, MGMT-methylated GBM [28]. The study did not meet its primary endpoint of OS in the overall randomized population or in patients with no baseline corticosteroid use [28].

Ineffectiveness of checkpoint inhibitor monotherapy in the recurrent GBM setting and combination of checkpoint inhibitor therapy with RT and temozolomide in the newly diagnosed GBM setting led to consideration of alternative approaches. One such approach is neo-adjuvant use of anti-PD-1 treatment where checkpoint inhibitors are administered preoperatively [29]. In a small study of 35 patients with recurrent GBM planned for repeat surgery, 19 patients were treated with anti-PD-1 agent pembrolizumab 14 ± 5 days prior to resection, then continued immunotherapy [29]. The remaining 16 patients started immunotherapy following resection [29]. mOS for patients receiving neoadjuvant pembrolizumab was 417 days compared to 228.5 days for patients receiving adjuvant pembrolizumab alone [29]. However, in a similar single-arm study where nivolumab was administered pre-operatively to patients with surgically resectable recurrent GBM, mOS was 7.3 months (approximately 220 days) [30]. Given the small numbers of participants, both results should be interpreted with caution and should not be generalized. It is also noted that in clinical practice, only a small subset of patients with GBM are eligible for additional surgery at the time of recurrence, typically representing smaller tumors, younger patients with better performance status, and greater initial resection [31].

Patient selection based on tumor mutational burden may represent another approach for successful use of immune checkpoint inhibitors for GBM treatment. In GBM, high tumor mutational burden can occur either due to the presence of DNA polymerase and mismatch repair defects within tumor cells or as a post-treatment phenomenon after administration of RT and temozolomide [1,32]. Clinical trials are evaluating the efficacy of immunotherapy with pembrolizumab (NCT02658279) and combination therapy with ipilimumab and nivolumab (NCT04145115) in patients with recurrent GBM with hypermutator pheonotype [33,34]. 

Combining immune checkpoint inhibitors with other treatment modalities represents another strategy to generate an antitumor immune response. Combination of checkpoint inhibitor therapy with other modalities such as hyperfractionated radiotherapy (NCT03532295; NCT03661723), laser interstitial thermal therapy (LITT) (NCT02311582; NCT03277638) is also being studied in clinical trials [35,36,37,38]. Ongoing checkpoint inhibitor trials are summarized in Table 1. Combination of these agents with other immunotherapy approaches that also leverage tumor molecular genetics is explored in the following sections.

To date, a majority of studies have focused on anti-PD-1/PD-L1 and anti-CTLA-4 approaches. Alternative checkpoint therapy targets include cluster of differentiation 47 (CD47) and cluster of differentiation 73 (CD73) [22]. CD47 binds signal-regulatory protein alpha (SIRPα) to inhibit macrophage-mediated phagocytosis [22]. Hu5F9-G4 is an anti-CD47 immunotherapy that has demonstrated preclinical activity against GBM, but no clinical trials have been completed to date for glioma treatment with this agent [39].

Extracellular adenosine is thought to have an immunosuppressive effect in the tumor microenvironment [22]. Adenosine monophosphate (AMP) is degraded to adenosine by CD73 [22]. Oleclumab, also known as MEDI9447, is an anti-CD73 antibody that was shown to prevent AMP-mediated lymphocyte suppression in preclinical models [40]. Combination of oleclumab with anti-PD-1 agent durvalumab is the subject of clinical trial NCT02503774 for advanced solid tumors [41]. Other monoclonal antibodies and small-molecule inhibitors of CD73 are being developed [22]. To date, no clinical trial data is available for GBM, but in a preclinical study, absence of CD73 improved survival in a murine model treated with anti-CTLA-4 and anti-PD-1 therapy [42]. Further studies are needed to determine the safety, feasibility, and efficacy of CD47 and CD73-based approaches for GBM treatment.

## 3. CAR T-Cell Therapy

CARs are synthetic receptors designed to direct T-cells to recognize and eliminate cells expressing a specific target antigen [43]. CARs generally consist of an extracellular antigen recognition domain and a transmembrane domain interacting with an intracellular T-cell signaling domain [43]. In CAR T-cell therapy, T lymphocytes collected from patients are modified by methodologies such as plasmid transfection or viral vector transduction to express a CAR, allowed to proliferate, and administered back to the patient with the goal of eliciting a durable tumor-specific immune response [43]. Multiple CAR T-cell products have been studied for treatment of GBM.

Overexpression of interleukin (IL)-13 receptor IL13Rα2 is observed in multiple types of cancer including an estimated >75% of GBMs [44]. IL13 signaling via activation of IL13Rα2 results in phosphoinositide 3-kinase (PIK3) pathway activation, promoting tumor cell proliferation [44]. In pilot study, three patients were treated with 12 intracavitary infusions of IL13Rα2-based CAR T-cell therapy with an mOS of 11 months [44]. The second generation of this CAR T-cell product was administered to a single patient with recurrent GBM and associated leptomeningeal disease, resulting in a complete response lasting 7.5 months [45]. However, recurrence after IL13Rα2-based therapy has been observed and optimization approaches and development of other CAR T-cell products are underway before wider-spread application can occur [45,46,47].

Receptor tyrosine-protein kinase ErbB2/HER2 expression in elevated in approximately 41% of GBM samples, representing another potential CAR T-Cell therapy target [48]. Safety concerns were raised with this approach due to death of one patient with colon cancer after receiving ErbB2-based CAR T-cell therapy following lymphodepleting chemotherapy [49]. A modified CAR T-cell product demonstrated safety but had limited T-cell persistence [50]. This led to exploration of virus-specific T cells as a means to deliver antitumor activity via ErbB2-based CAR while receiving costimulation by latent virus antigens presented by APCs [48]. In a phase I clinical trial, 17 patients with recurrent GBM were treated with Epstein–Barr virus (EBV), adenovirus, and cytomegalovirus (CMV)-specific T cells that also recognized ErbB2 [48]. Treatment was well-tolerated with an mOS of 11 months reported [48].

Epidermal growth factor receptor (EGFR) amplification is frequently encountered in newly diagnosed GBM [51]. EGFR variant III (EGFRvIII) is the most common EGFR alteration found in GBM, representing an in-frame deletion from exons 2–7 causing an extracellular domain alteration resulting in constitutive activation [51]. EGFRvIII-targeting CAR T-cell therapy was studied in 10 patients with recurrent GBM, administered as a peripheral infusion with mOS of 8 months, though this was a heavily pre-treated patient population [52].

Use of naturally occurring tumor-binding molecules represents an alternative method of developing CAR T-cell therapy [53]. Chlorotoxin (CLTX) is one such molecule with GBM-binding potential, derived from the venom of *Leiurus quinquestriatus* [53,54]. In preclinical models, CLTX CAR T cells demonstrated anti-tumor activity with a clinical trial underway (NCT04214392) [53,55]. 

The disialoganglioside GD2 is highly expressed on target H3K27M-mutated glioma cells. H3K27M mutation is commonly found on diffuse midline gliomas, including Diffuse Intrinsic Pontine Glioma (DIPG) [56]. There was concern that the use of immunotherapy in the treatment of DIPG would be precarious and result in a lethal rhombencephalitis in this patient population [56]. Increased intracranial pressure and brainstem edema as a consequence of obstructive hydrocephalus can be life-threatening unless immediately and appropriate managed [56]. However, in their recent report, Monje et al. described that the toxicities associated with the GD2 CAR T-cell infusions were manageable and reversible with supportive care [56]. There was no obvious sign or symptom of off-tumor, off-target toxicity involving the brain, or peripheral nerves, as can be seen with other anti-GD2 antibodies for the treatment of neuroblastoma [56]. Notably, cytokine release syndrome and immune effector cell-associated neurotoxicity were in line with prior reports of CAR T therapy and no worse [56]. Three of these four patients treated derived radiographic and clinical benefit after intravenous administration of anti-GD2 CAR T cells [56]. 

CAR T approaches in glioma are summarized in Table 2. While single-antigen CAR T cells may create an anti-tumor immune response expressing that antigen, the target may not be present in all tumor cells [1,57]. Epitope spreading refers to the concept of diversifying epitope specificity and targeting multiple areas of a single protein or multiple tumoral proteins [57]. Application of this concept by developing CAR T-cell products targeting multiple antigens may result in a more robust whole-tumor immune response. Combination of CAR T-cell therapy with immune checkpoint inhibitors may similarly improve efficacy and duration of anti-tumor immunity.

## 4. Vaccine-Based Therapy

Vaccine-based therapies are intended to elicit an anti-tumor response by introducing T cells to immunogenic tumor-specific antigens unique to tumor cells or tumor-associated antigens overexpressed on tumor cells [1,11]. While identification of target antigens sufficiently expressed in the entire population of tumor cells and the phenomenon of antigen escape represent significant challenges, multiple clinical trials already completed in this arena have added greatly to the understanding of GBM immunotherapy [1].

Rindopepimut is an EGFRvIII-targeting peptide vaccine that was studied in a phase III, randomized, double-blind, clinical trial for patients with newly diagnosed GBM with confirmed EGFRvIII expression by central analysis who had undergone maximal resection and standard-of-care radiation therapy with concomitant temozolomide [58]. Patients were randomized to monthly intradermal vaccine injections compared to control, concurrent with adjuvant oral temozolomide [58]. A total of 745 patients were enrolled; 371 received vaccine treatment and 374 received control. The study was terminated for futility with no difference in mOS between the two groups [51]. Importantly, loss of EGFRvIII expression was described ~57–59% of tumors in both treatment and control arms [59]. Loss of EGFRvIII expression was not correlated with vaccine treatment or anti-EGFRvIII antibody titers, suggesting a change occurring with GBM progression rather than a response to vaccine treatment [59].

Survivin is an intracellular anti-apoptotic protein that inhibits caspase activation and has a role in regulation of cell division [60]. Survivin is highly expressed in GBM cells, representing a potential vaccine target [60]. In a study of nine patients with recurrent GBM, survivin peptide mimic SurVaxM was associated with mOS of 86.6 weeks [60]. Ongoing clinical trial NCT04013672 is evaluating SurVaxM in combination with pembrolizumab in patients with recurrent GBM [61].

Wilm’s tumor 1 (WT1) is a transcription factor involved in oncogenesis and detected in solid organ tumors including GBM [62]. In a phase I study, 24 patients with advanced solid organ tumors were treated with WT1 peptide vaccine DSP-7888 [62]. Intradermal and subcutaneous routes of administration were compared with higher WT1-specific cytotoxic lymphocyte induction noted with intradermal injection [62]. Seven patients with GBM were included in the study, two of whom had stable disease [62]. Ongoing clinical trial NCT03149003 is evaluating DSP-7888 in combination with bevacizumab for patients with recurrent GBM [63].

VXM01 is a plasmid containing an attenuated *Salmonella typhi*, TY21a that encodes vascular endothelial growth factor receptor-2 (VEGFR-2) [64]. Administration of vaccine platform VXM01 is intended to recruit VEGFR-2-targeting T cells to target the tumor and its vasculature [64]. In early data presented, 14 patients with progressive GBM were treated with the vaccine with a decrease in intratumoral PD-L1 expression correlated with increased survival [64]. Thus, in ongoing clinical trial NCT03750071 for patients with recurrent GBM, VXM01 is combined with anti-PD-L1 checkpoint inhibitor avelumab [65].

IMA950 is a multipeptide vaccine developed based on antigen expression patterns on the surface of GBM samples [66]. IMA950 contains nine major histocompatibility complex (MHC) class I and two MHC class II peptides [66]. Poly-ICLC is an adjuvant administered with the vaccine that was shown to enhance vaccination efficacy in mouse glioma model [66]. In a phase I/II clinical trial, patients with newly diagnosed GBM were treated with the vaccine [66]. The first six patients received IMA950 intradermally and poly ICLC intramuscularly [66]. Vaccine-induced CD8+ T-cell responses were restricted to a single peptide and CD4+ T-cell responses were absent [66]. After protocol amendment, IMA950 and poly-ICLC were mixed and 13 additional patients were treated with 63.2% single-peptide, 36.8% multi-peptide CD8+ T-cell responses as well as 84.6% of patients had tumor-peptide specific CD4+ T-cell responses [66]. mOS was 19 months [66]. Ongoing clinical trial NCT03665545 is studying IMA950/Poly-ICLC in combination with pembrolizumab for recurrent GBM [67].

Another multipeptide vaccine under investigation is EO2401, which consists of three ‘oncomimics’, described as peptides homologous, but not identical, to tumor antigens [68,69]. Ongoing clinical trial NCT04187404 is studying use of EO2401 in metastatic adrenocortical carcinoma as well as malignant pheochromocytoma and paraganglioma [68]. In a parallel phase 1b/2a clinical trial NCT04116658, immunogenicity of the vaccine is being assessed in the setting of recurrent GBM [69].

Personalized neoantigen vaccines represent a different approach to anti-tumor vaccine development informed by sequencing data from individual tumors. In a phase I/Ib trial, a neoantigen vaccine was administered to 10 patients with newly diagnosed GBM [70]. mOS of 16.8 months was reported with neoantigen-specific CD4+ and CD8+ T-cell response noted in patients who were not receiving dexamethasone, suggesting potential for neoantigen targeting vaccines to alter the GBM immune milieu in absence of corticosteroid treatment [70]. 

In another phase I trial, 15 patients with newly diagnosed GBM were treated with personalized vaccine APVAC1, derived from premanufactured library of antigens based on tumor sequencing as well as APVAC2, targeting neoepitopes [71]. Sustained CD8+ memory T-cell responses were noted with APVAC1 and predominantly CD4+ responses were noted with APVAC2 [71]. Ongoing clinical trial NCT02287428 is evaluating use of a personalized neoantigen vaccine in combination with pembrolizumab for patients with newly diagnosed GBM [72].

An alternative approach to antitumor vaccine development for GBM involves use of antigen-presenting dendritic cells (DCs) to generate a polyvalent immune response [11]. In a phase II, double-blind, placebo-controlled randomized clinical trial for patients with newly diagnosed GBM, ICT-107 DC vaccine was studied [73]. GBM stem cell-associated peptides MAGE-1, HER-2, AIM-2, TRP-2, gp100, and IL13Rα2 were selected as antigens of interest for development of this DC vaccine [73]. Autologous monocytes were harvested from each subject with DC differentiation stimulated by culturing in recombinant granulocyte monocyte colony stimulating factor (GM-CSF) and interleukin 4 (IL4), followed by interferon gamma and lipopolysaccharide [73]. Resulting DCs were incubated with 9–10 amino acid synthetic peptides derived from the six pre-selected antigens [73]. Subjects were subsequently administered 1 mL of pulsed DCs (1.1 × 10^7^ cells/mL) with controls administered 1 mL of DCs unpulsed with antigen (3.6 × 10^6^ cells/mL) [73]. A total of 124 patients were enrolled, randomized 2:1 between ICT-107 and unpulsed DC [73]. mOS was 17 months in the treatment group compared to 15 months in the control group, which was not statistically significant [73]. Further analysis revealed a differential immune response based on HLA status [73]. MAGE-1 and AIM-2 were selected as HLA-A1 antigens. gp100, HER2/neu, IL13Rα2, and TRP-2 were selected as HLA-A2 antigens [73]. Though not statistically significant, the mOS benefit only occurred in HLA-A2 patients independent of MGMT methylation status [73]. This observation suggests a potential role for antigen and patient selection based on HLA typing in future studies.

In a phase II randomized trial in patients with newly diagnosed GBM with at least 70% resection, patients were randomized in a 1:1 fashion to standard-of-care therapy (SOC) and SCO alongside dendritic cell vaccine Audencel [74]. Vaccine was administered weekly during weeks 7 to 10, followed by monthly intervals [74]. mOS was 18.3 months in both groups [74].

In another study conducted at China Medical University, 34 patients with newly diagnosed GBM were treated with SOC therapy with or without adjuvant autologous dendritic cell vaccine [75]. Vaccination was administered starting 1–2 months postoperatively and continued over a 6-month period [75]. Among the 76 patients studied, mOS for the vaccine group was 31.9 months, comparing favorably to the control group at 15.0 months [75]. However, Results of the ICT-107 DC vaccine, Audencel DC vaccine, and China Medical University DC vaccine studies were combined in a metanalysis and no substantial effect on mOS was noted [76].

In a different study, autologous DCs were pulsed with tumor cell lysate [77]. 331 patients with newly diagnosed GBM were randomized 2:1 between DCVaxL and placebo administered following completion of radiation therapy during treatment with adjuvant temozolomide on days 0, 10, and 20, then months 2, 4, and 8, and thereafter at 6-month intervals starting at month 12 [77]. Of note, 1599 patients were screened for the study with 1268 excluded for reasons such as non-GBM diagnosis (306), insufficient tumor lysate (201), disease progression (250), issues with vaccine manufacture (75), and unsuccessful leukapheresis (61) [77]. Among 331 evaluable patients, 232 were randomized to vaccine and 99 to placebo with crossover permitted at progression and 90% of participants ultimately receiving vaccine treatment [77]. Preliminary data indicated 23.1 months mOS, which compares favorably to historical mOS of 15–17 months from other studies with further analysis pending [77]. Ongoing clinical trial NCT04201873 is evaluating dendritic cell vaccine ATL-DC with checkpoint inhibitor pembrolizumab in patients with surgically accessible recurrent GBM [78].

Another autologous dendritic cell vaccine was studied in a phase II clinical trial in patients with newly diagnosed GBM undergoing fluorescence-guided maximal resection with less than 1 cm^3^ residual tumor [79]. Following surgery, patients were treated with radiation therapy with concomitant temozolomide chemotherapy followed by up to 12 cycles of adjuvant temozolomide or until disease progression [79]. Dendritic cell vaccine was administered prior to radiation therapy, second three weeks after radiotherapy, followed by two monthly, four bi-monthly, and subsequent quarterly administrations [79]. Among 32 evaluable patients, mOS was 23.4 months [79]. In a phase I/II study, autologous dendritic cell vaccine was studied in 77 patients with newly diagnosed GBM [80]. Following radiation therapy, patients received four weekly vaccine administrations [80]. Four additional vaccinations were administered during adjuvant temozolomide chemotherapy [80]. Among evaluable patients, mOS was 18.3 months [80]. 

Vaccine-based therapies in glioblastoma are summarized in Table 3. To date there have been no major successes with use of single-peptide, multipeptide, neoantigen-based, or dendritic cell-based vaccines in GBM leading to broad implementation. However, prior studies have provided key findings to assist future vaccine development such as loss of EGFRvIII expression noted with rindopepimut highlighting the phenomenon of antigen escape, the importance of PD-L1 expression noted with VXM01, and HLA-based response noted for ICT-107 that can inform target antigen selection, patient selection, and multi-agent combinatorial approaches in future studies [59,64,73]. Despite good tolerability and feasibility demonstrated across multiple studies, DC vaccine approaches did not demonstrate a survival benefit in a metanalysis [76]. Given significant differences in administration schedules utilized in DC vaccine studies to date, optimization of vaccination timing and patient selection based on HLA profiles may improve results. With multiple ongoing studies, development of vaccine-based strategies for GBM treatment remains an active area of research.

## 5. Viral Therapies

Oncolytic viruses can be administered either intravenously or intratumorally to be selectively taken up by tumor cells, generating an initial cytotoxic response with intention to trigger antigen presentation to the immune system and resulting in durable adaptive and innate immune response [81]. Multiple viral vectors have been studied for treatment of GBM, attempting to elicit durable antitumor immune responses [1].

AdV-tk is an adenoviral vector that has been studied in a phase II clinical trial in patients with newly diagnosed GBM, anaplastic astrocytoma, and anaplastic oligodendroglioma [82]. AdV-tk, which contains the herpes simplex virus thymidine kinase gene, was administered to the resection bed during surgery and selectively taken up by rapidly dividing tumor cells rather than quiescent nearby normal brain tissue [82]. Patients were subsequently treated with valacyclovir, which competitively inhibits DNA synthesis in infected cells, resulting in cell death [82]. Among GBM patients treated in the study, mOS was 16.7 months, compared to 13.7 months in controls, favoring the treatment arm with the greatest benefit observed in patients undergoing a gross total resection [82]. In an ongoing clinical trial NCT03576612 for patients with newly-diagnosed high grade gliomas, AdV-tk intratumoral injection followed by valacyclovir is being combined with checkpoint inhibitor nivolumab for additional antitumor immune response [83].

Interleukin 12 (IL-12) is a cytokine with anticancer activity, but limited application as a systemic therapeutic agent due to severe toxicity [84]. A ligand-inducible expression switch called the RheoSwitch Therapeutic System^®^ (RTS^®^) was designed for local control of IL-12 production within the tumor microenvironment [85]. The system relies on use of activator ligand velemidex with delivery of IL-12 transgene achieved by use of an adenoviral vector Ad-RTS-hIL-12 [85]. In a phase I study of 31 patients with recurrent glioma, the safety and tolerability of this approach was established [86]. In a subsequent open-label phase I dose-escalation trial, use of Ad-RTS-hIL-12, velemidex, and checkpoint inhibitor nivolumab was studied as a combinatorial immunotherapy with tolerable toxicity [85]. A larger phase II study NCT04006119 is in progress for combination of Ad-RTS-hIL-12, velemidex, and anti-PD-1 checkpoint inhibitor cemiplimab [87].

Local IL-12 release is also leveraged with oncolytic herpes simplex virus (oHSV) M032 [88,89]. The oncolytic virus is selectively taken up by tumor cells resulting in an initial cytotoxic effect while simultaneously causing tumor cells to secrete IL-12 [88]. Tolerability of the approach was studied in twenty-five canine patients [89]. Results from NCT02062827, a phase I study of M032 in patients with recurrent glioma are pending [90]. An ongoing phase II clinical trial NCT05084430 is evaluating combination of M032 with pembrolizumab in patients with recurrent glioma [91].

Another oHSV viral vector under study for treatment of recurrent GBM is rQNestin34.5v.2 [92]. This vector is modified to replicate only in GBM cells that express nestin, which is a stem cell marker [92]. Ongoing clinical trial NCT03152318 is studying intravenous administration of cyclophosphamide preoperatively followed by intratumoral administration of rQNestin34.5v.2 [93]. In this setting, chemotherapy agent cyclophosphamide is intended to have an immunomodulatory effect [94].

oHSV G47Δ similarly selectively replicates in cancer cells, including GBM stem cells [95,96]. In a phase II clinical trial, G47Δ was administered by stereotactic intratumoral injection to patients with recurrent GBM up to six times [95]. Repeated stereotactic injections were well tolerated with one-year survival among 13 patients reported at 92% with additional studies of the oHSV planned [95].

DNX-2401 is an oncolytic adenovirus modified to replicate selectively in retinoblastoma pathway-deficient cells and intended to elicit tumor necrosis while subsequently eliciting an anti-tumor immune response [97,98]. In a phase I study, 31 patients with recurrent GBM were treated with DNX-2401 [98]. Group I received intratumoral injection without resection [98]. Group II underwent injection followed by surgical resection and a second injection 14 days later [98]. Patients in the resection group had mOS of 13.5 months compared to 9.5 in the non-resection group with concurrent corticosteroid use identified as a factor negatively influencing survival [98]. Ongoing trial NCT03896568 is evaluating intra-arterial injection of the oncolytic adenovirus in patients with recurrent high-grade glioma [99].

CRAd-Survivin-pk7 is another oncolytic adenovirus under evaluation for GBM [100]. In a phase I clinical trial, HB1.F3-CD human neural stem cells were loaded with the oncolytic virus and injected into walls of the resection cavity after surgery with intended selective uptake of the vector by tumor cells [100]. Twelve patients with newly diagnosed high-grade glioma received injections followed by standard-of-care radiation therapy with concomitant and adjuvant temozolomide chemotherapy [100]. mOS was 18.4 months with no dose-limiting toxicities encountered and increase in CD8+ T cells noted at the highest studied dose [100]. Larger scale clinical trials are planned.

PVSRIPO is a live attenuated poliovirus type 1 vaccine under study for antitumor effect in GBM as well as melanoma [101,102]. This oncolytic virus has an internal ribosome entry site replaced with that of human rhinovirus type 2, which ablates neurovirulence [103]. CD155, which is broadly upregulated in malignant cells, mediates virus tropism for tumor [101]. Similar to other oncolytic viruses, the intent is to induce a cytotoxic effect followed by antitumor immune response [103]. In a phase I clinical trial for patients with recurrent GBM, 61 patients were enrolled and received vaccine intratumorally [101]. During the dose expansion phase of the trial, overall survival was 21% at 24 months with 19% of the participants experiencing PVSRIPO-related grade 3 or higher adverse events such as seizure, confusion, or pyramidal track syndrome [101]. Further results from phase II clinical trial NCT02986178 are pending [104].

A different approach to viral-based tumor treatment strategies involves use of gene therapy where viruses rendered replication-incompetent are administered to deliver anticancer complementary DNA (cDNA) [1]. Vocimagene amiretrorepvec (Toca 511) is a gamma-retroviral replicating vector that encodes cytosine deaminase [105]. Following injection of the virus into the resection wall, treatment with extended-release 5-fluorocytosine (Toca FC) leads to local production of 5-fluorouracil, which depletes immuno-suppressive myeloid cells and helps induce antitumor immunity [105]. A 21.7% durable response rate was reported in an early phase trial with use of Toca 511 and Toca FC [105]. In a subsequent phase III trial, 403 patients with recurrent GBM were randomized 1:1 between viral therapy and standard-of-care chemotherapy [106]. Primary endpoint was not met with mOS 11.1 months for the treatment arm compared to 12.2 months for the control arm [106]. It has been noted that median prodrug dosing was suboptimal in the phase III study and further investigation may identify specific patient populations who may benefit [107].

Viral therapies in glioma are summarized in Table 4. These therapies have so far had limited application in GBM with early phase clinical trials recruiting small numbers of patients [1]. Intratumoral injection does limit application to those patients who are candidates for additional surgery at the time of GBM recurrence. However, the dual delivery of an immediate cytotoxic effect followed by antitumor immunity may eventually become a viable option in the newly diagnosed GBM setting.

## 6. Cytokine Therapy

Standard of care for radiation therapy and temozolomide used for management of GBM is associated with severe prolonged lymphopenia in about 40% of patients and is associated with poorer patient survival [108]. A recent study showed that a novel long-acting interleukin-7 agonist, NT-I7, demonstrated the ability to correct this treatment-related lymphopenia and significantly increased peritumoral CD8 lymphocytes and improved survival in murine GBM models [109].

Clinical trials are ongoing to evaluate the effect of NT-I7 on lymphocytes and survival in patients with high-grade gliomas. Early data from NCT03687957 showed that NT-I7 increased lymphocyte numbers with minimal toxicity [110]. Further study is needed to determine whether there is survival benefit of this approach in patients with GBM.

## 7. Future Directions

Successful implementation of immunotherapy in GBM remains elusive. Negative phase III clinical trials with checkpoint inhibitors, vaccine-based therapy rindopepimut, and viral therapy Toca 511 are disappointing [26,27,28,58,106]. However, studies for multiple additional vaccine- and viral-based therapies are underway along with CAR T cell-based therapy and combination approaches using checkpoint inhibitors. Future larger scale implementation of immunotherapy in GBM remains possible. Approaches will need to consider issues including antigen escape, tumor heterogeneity, tumor microenvironment, drug delivery strategies, patient selection informed by tumor genetics, and multimodality treatment approaches. 

## Figures and Tables

**Table 1 ijms-23-07046-t001:** Ongoing Checkpoint Inhibitor Clinical Trials in Glioblastoma.

Trial Identifier	Title	Phase	Tumor Type
NCT02658279	Pembrolizumab (MK-3475) in Patients with Recurrent Malignant Glioma with a Hypermutator Phenotype	N/A ^1^	Recurrent Glioma
NCT04145115	A Study Testing the Effect of Immunotherapy (Ipilimumab and Nivolumab) in Patients with Recurrent Glioblastoma with Elevated Mutational Burden	2	Recurrent GBM ^2^
NCT03532295	Retifanlimab and Epacadostat in Combination with Radiation and Bevacizumab in Patients with Recurrent Gliomas	2	Recurrent Glioma
NCT03661723	Pembrolizumab and Reirradiation in Bevacizumab Naïve and Bevacizumab Resistant Recurrent Glioblastoma	2	Recurrent GBM
NCT02311582	MK-3475 in Combination with MRI-Guided Laser Ablation in Recurrent Malignant Gliomas	1/2	Recurrent Glioma
NCT03277638	Laser Interstitial Thermotherapy (LITT) Combined with CheckPoint Inhibitor for Recurrent GBM (RGBM)	1/2	Recurrent GBM

^1^ Study phase not listed on ClinicalTrials.gov; ^2^ GBM: Glioblastoma.

**Table 2 ijms-23-07046-t002:** Chimeric Antigen Receptor (CAR) T-Cell Therapies in Glioma.

Target	Summary
IL13Rα2	(IL)-13 receptor frequently overexpressed in GBM. In a pilot study, 3 patients with rGBM ^1^ treated, mOS ^2^ 11 months.
ErbB2/HER2	Receptor tyrosine kinase ErbB2/HER2 frequently overexpressed in GBM. In a phase I study, 17 patients with rGBM treated, mOS 11 months.
EGFRvIII	Epidermal growth factor receptor (EGFR) variant III (EGFRvIII) is the most common EGFR alteration found in GBM. In a phase I study, 10 patients with rGBM treated, mOS 8 months.
CLTX	Chlorotoxin (CLTX) is a naturally occurring tumor-binding molecule. Phase I clinical trial NCT04214392 underway.
GD2	Disialoganglioside GD2 is highly expressed on target H3K27M-mutated glioma cells. In a phase I study, 4 patients with H3K27M mutant midline glioma treated, 3 had radiographic response.

^1^ rGBM: Recurrent glioblastoma; ^2^ mOS: Median overall survival.

**Table 3 ijms-23-07046-t003:** Vaccine-Based Therapies in Glioblastoma.

Target/Product	Summary
EGFRvIII Rindopepimut	Rindopepimut is an EGFRvIII targeting peptide vaccine. Phase III, randomized, double-blind, clinical trial for newly diagnosed GBM ^1^ with EGFRvIII expression terminated for futility due to no difference in mOS ^2^
Survivin SurVaxM	Survivin is an intracellular anti-apoptotic protein highly expressed in GBM cells. In a phase I study, 9 patients with rGBM ^3^, treated with survivin peptide mimic SurVaxM, mOS 86.6 weeks. Phase II clinical trial NCT04013672 underway.
WT1 DSP-7888	Wilm’s tumor 1 (WT1) is a transcription factor detected in GBM. In a phase I study, 7 patients with rGBM treated with WT1 peptide vaccine DSP-7888, 2 had stable disease. Phase III clinical trial NCT03149003 underway.
VEGFR-2 VXM01	Vaccine platform VXM01 includes an attenuated Salmonella typhi, TY21a that encodes vascular endothelial growth factor receptor-2 (VEGFR-2). In a phase I study, 14 patients with rGBM, treated with tolerable safety. Phase I/II clinical trial NCT03750071 underway.
Multipeptide IMA950	IMA950, multipeptide vaccine based on GBM cell surface antigen expression patterns. In a phase I/II study, 19 patients with GBM treated, mOS 19 months after protocol amendment and product modification. Phase I/II clinical trial NCT03665545 is underway.
Multipeptide EO2401	EO2401, multipeptide vaccine based on oncomimics. Phase Ib/IIa clinical trial NCT04116658 is underway.
Personalized Neoantigen Vaccine	In a phase I/Ib trial, APVAC neoantigen vaccine administered to 10 patients with new GBM, mOS of 16.8 months.
Personalized APVAC	In a phase I trial, personalized vaccine APVAC administered to 15 patients with new GBM, tolerable safety.
Dendritic Cell ICT-107	Uses antigen-presenting dendritic cells (DCs) to generate a polyvalent immune response. In a phase II, randomized, placebo-controlled trial, 124 patients treated, mOS 17 months in treatment group compared to 15 months in placebo group.
Dendritic Cell Audencel	In a phase II, 1:1 randomized trial, 76 patients treated, mOS 18.3 months in both treatment and control groups.
Dendritic Cell China Medical University trial	In a phase II, randomized trial, 34 patients treated, mOS 31.9 months in treatment group compared to 15 months in placebo group.
Dendritic Cell DCVaxL	In a phase III, randomized, placebo-controlled trial, 331 patients treated, mOS 23.1 months in treatment group compared to historical mOS 15–17 months, further analysis pending.
Dendritic Cell EudraCT 2009-009879-35 trial	In a phase II trial, 32 patients with new GBM who underwent fluorescence-guided maximal resection treated, mOS 23.4 months.
Dendritic Cell EudraCT 2006-002881-20 trial	In a phase I/II trial, 77 patients with new GBM treated postoperatively, mOS 18.3 months.

^1^ GBM: Glioblastoma; ^2^ mOS: Median overall survival; ^3^ rGBM: Recurrent glioblastoma.

**Table 4 ijms-23-07046-t004:** Viral Therapies in Glioma.

Product	Summary
AdV-tk	Adenoviral vector. In a clinical trial, mOS ^1^ was 16.7 months in GBM ^2^ patients. Phase I clinical trial NCT03576612 underway.
Ad-RTS-hIL-12	Adenoviral vector that uses activator ligand velemidex for delivery of IL-12 transgene. In a phase I study, 31 patients with recurrent glioma treated, tolerable safety. Phase II clinical trial NCT04006119 ongoing.
M032	Oncolytic herpes simplex virus (oHSV). Phase I clinical trial results pending. Phase II clinical trial NCT05084430 ongoing.
rQNes-tin34.5v.2	oHSV viral vector. Phase I clinical trial NCT03152318 ongoing.
oHSV G47Δ	oHSV viral vector. In a phase II clinical trial, survival among 13 patients reported at 92%.
DNX-2401	Oncolytic adenovirus modified to replicate in retinoblastoma pathway deficient cells. In a phase I study, 17 patients with rGBM ^3^ treated, mOS of 13.5 months compared to 9.5 in the non-resection group.
CRAd-Survivin-pk7	Oncolytic virus. In a phase I study, 31 patients with new GBM treated, mOS 18.4 months.
PVSRIPO	Live attenuated poliovirus. Internal ribosome entry site replaced with that of human rhinovirus type 2, which ablates neurovirulence. In a phase I clinical trial, 61 patients enrolled, overall survival 21% at 24 months. Phase II clinical trial NCT02986178 underway.
Toca 511	Gamma-retroviral replicating vector that encodes cytosine deaminase. In a phase III trial, 403 patients with rGBM treated, mOS 11.1 months compared to 12.2 months for the control arm.

^1^ mOS: Median overall survival; ^2^ GBM: Glioblastoma; ^3^ rGBM: Recurrent glioblastoma.

## Data Availability

Data availability is not applicable to this review article.

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
