# Peer review of "Immunotherapy in Glioblastoma: Current Approaches and Future Perspectives"

_ijms, 2022, doi:10.3390/ijms23137046_

Round 1
Reviewer 1 Report
This is a very well written review article that puts together all facets of immunotherapy and how they relate to glioblastoma treatment. The authors go into great detail regarding checkpoint inhibitors, CAR t cell therapy, vaccine based therapies, viral therapies, and cytokine therapy, as well as currently open studies.
I have no edits--the authors presented the topics well organized and in longitudinal fashion.
Author Response
Reviewer #1: This is a very well written review article that puts together all facets of immunotherapy and how they relate to glioblastoma treatment. The authors go into great detail regarding checkpoint inhibitors, CAR t cell therapy, vaccine based therapies, viral therapies, and cytokine therapy, as well as currently open studies. I have no edits--the authors presented the topics well organized and in longitudinal fashion.
[Authors]: Thank you very much for your kind review of our article.
Reviewer 2 Report
Title
Immunotherapy in Glioblastoma: Current Approaches and Future Perspectives.
Concise Summary
The authors review the immunotherapy approaches to glioblastoma (GBM). that there are promising results about vaccine and viral-based therapies with from CAR cell based therapy and combined treatment using check-point inhibitors with temozolomide or radiotherapy. They conclude implementation of immunotherapy in GBM should evaluate other important issues as tumor heterogeneity, tumor microenvironment, drug delivery strategies and multimodality treatment approaches.
Major criticism
1.The manuscript has been written correctly, and it describes the topic presented in some detail. However, the study does not provide any news nor is the perspective of the treatment of the GBM with respect to what has been published (PMID: 32253714. PMID: 34054867. PMID: 35428347). The most informative aspect of the work is the table display of a list of either completed or ongoing projects, but the conclusions do not provide anything new and are very general.
2. The largest contribution of the manuscript can be summarized in the presentation of the ongoing or recently finalized studies, but there is no assessment of the relevance of these results. This approach would have been much more interesting for the reader. Moreover, meta-analyses that question the success of the vaccines have not been considered. For example, PMID: 34767325.
3.The review of published studies of dendritic cell vaccines is incomplete as there are some relevant studies that have not been considered as PMID: 28499389.
4.In the title of the work it is indicated that the issue "future perspectives" will be dealt with, however no new ideas are provided on what has already been indicated in previous reviews.
Minor criticism
1.In addition to not being real, the accuracy of the 5-year overall survival at 7.2% is somewhat ridiculous to illustrate the severity of the disease (line 10).
2. Sentences such as “is also being studied in clinical trials with promising signals[32-35]” should not be cited based on studies whose definitive results have not been published (line 115). In addition, the word “promising” repeated several times in the article means nothing.
3. The eponym “Epstein-Barr virus” is misspelled in the phrase “patients with recurrent GBM were treated with Ebstein-Barr virus (EBV), adenovirus … (line 151).
4. The authors include unpublished studies in the bibliography, dated in 2023 and later years. These are work-in-progress citations that are not necessary to include until results are available. (Washington University School of, M. and C. Incyte, Retifanlimab and Epacadostat in Combination With Radiation and Bevacizumab in Patients With Recurrent Gliomas. 2025. Washington University School of, M., S. Merck, and L.L.C. Dohme, MK-3475 in Combination With MRI-guided Laser Ablation in Recurrent Malignant Gliomas. 2024. (Lines 512-13;533-36).
Conclusion
Finally, I consider that it is an interesting review, that provides practical information about this item. However, it does not go give relevant information on previous retrospective surveys on this issue. Several suggestions have been provided to the authors to improve the quality of the article.
Author Response
Major criticism
1.The manuscript has been written correctly, and it describes the topic presented in some detail. However, the study does not provide any news nor is the perspective of the treatment of the GBM with respect to what has been published (PMID: 32253714. PMID: 34054867. PMID: 35428347). The most informative aspect of the work is the table display of a list of either completed or ongoing projects, but the conclusions do not provide anything new and are very general.
[Authors]: We have included references to the abovementioned review articles. Additionally, based on the article from Rong et al, we made additional changes to expand the provided literature review:
- Included paragraph on CD47 targeting checkpoint inhibitor therapy.
- Included paragraph on CD73-based checkpoint inhibitor therapies.
We note that this is an invited review article for a special issue of the journal titled ‘Recapitulating the Key Breakthroughs and Future Perspective’. Our goal with the article is to provide an overview of immunotherapy approaches that have been tried to date, thus recapitulating what has been learned and highlighting approaches that will be the subject of larger scale trials in the future. This article is not original research. It is not intended to provide new information, rather a reference point particularly for the uninitiated and starting point for further reading.
- The largest contribution of the manuscript can be summarized in the presentation of the ongoing or recently finalized studies, but there is no assessment of the relevance of these results. This approach would have been much more interesting for the reader. Moreover, meta-analyses that question the success of the vaccines have not been considered. For example, PMID: 34767325.
We included results from 30301187, 22120301, EudraCT 2009-009879-35, and EudraCT 2006-002881-20 to expand the dendritic vaccine section. We also included the metanalysis mentioned. The intention is to provide an overview of vaccination approaches. We expanded the conclusion of the vaccine section to further discuss relevance of the results.
3.The review of published studies of dendritic cell vaccines is incomplete as there are some relevant studies that have not been considered as PMID: 28499389.
As noted above, we included results from 30301187, 22120301, EudraCT 2009-009879-35, and EudraCT 2006-002881-20 to expand this section.
4.In the title of the work it is indicated that the issue “future perspectives” will be dealt with, however no new ideas are provided on what has already been indicated in previous reviews.
[Authors:] Ongoing studies and results of early phase studies are discussed throughout the paper. Novel approaches under investigation include combination of checkpoint inhibitors with other modalities of treatment, personalized vaccines where only early phase clinical studies are available, viral vector therapies where only early phase results are published, and CAR T treatments moving to larger phase trials. Thus, future directions of the field are represented throughout the manuscript. Based on above feedback we have also included data on CD47 and CD73 based strategies. We underscored issues of antigen escape, tumor heterogeneity, and immunosuppressive nature of GBM representing challenges that will need to be overcome. The intent of the article is not to propose novel ideas, but to capture the current state and where studies are headed. This has been now made more comprehensive with revisions discussed above.
Minor criticism
1.In addition to not being real, the accuracy of the 5-year overall survival at 7.2% is somewhat ridiculous to illustrate the severity of the disease (line 10).
[Authors:] We would like to direct the reviewer to the referenced article for the provided statistic, which has been reported by CBTRUS. Direct quote from the publication states: “Relative survival estimates for glioblastoma were quite low; 7.2% of patients survived five years post-diagnosis” (Ostrom et al, PMC7596247). This is in line with the previous edition of published CBTRUS data: “Five-year relative survival was lowest for glioblastoma (6.8%)” (Ostrom et al, PMC6823730). This was also cited by the comprehensive glioblastoma review by Wen et al with the following sentence: “Glioblastomas contribute disproportionately to morbidity and mortality, with a 5-year overall relative survival of only 6.8%, which varies by age at diagnosis and by sex (Fig. 1B; National Program of Cancer Registries, 2012–2016)” (Wen et al, PMC7594557). In the EF14 clinical trial for tumor treating fields, 5-year overall survival was reported notably higher at 13%, though we used the CBTRUS data in our manuscript as more representative of real world data (29260225). We added from CBTRUS data that median survival is 8 months to further communicate the severity of this malignancy (PMC7596247).
- Sentences such as “is also being studied in clinical trials with promising signals[32-35]” should not be cited based on studies whose definitive results have not been published (line 115). In addition, the word “promising” repeated several times in the article means nothing.
[Authors:] We have edited the article to simply point out studies are ongoing and removed the word ‘promising’.
- The eponym “Epstein-Barr virus” is misspelled in the phrase “patients with recurrent GBM were treated with Ebstein-Barr virus (EBV), adenovirus … (line 151).
[Authors:] We have corrected this.
- The authors include unpublished studies in the bibliography, dated in 2023 and later years. These are work-in-progress citations that are not necessary to include until results are available. (Washington University School of, M. and C. Incyte, Retifanlimab and Epacadostat in Combination With Radiation and Bevacizumab in Patients With Recurrent Gliomas. 2025. Washington University School of, M., S. Merck, and L.L.C. Dohme, MK-3475 in Combination With MRI-guided Laser Ablation in Recurrent Malignant Gliomas. 2024. (Lines 512-13;533-36).
[Authors:] We modified the Cytokine Therapy section to include only the study where data has already been published.
Conclusion
Finally, I consider that it is an interesting review, that provides practical information about this item. However, it does not go give relevant information on previous retrospective surveys on this issue. Several suggestions have been provided to the authors to improve the quality of the article.
[Authors:] Thank you very much for your kind and thorough review. We have made modifications to address the specific concerns addressed above, which includes adding additional references to address the concern mentioned here.
Round 2
Reviewer 2 Report
I appreciate the modifications that the authors brought to their manuscript according to the reviewers´ comments. The authors have made several changes in the manuscript, which has resulted in an improved new version of the article. The reviewer's observation about glioblastoma patient survival may not have been fully explained, but punctuating patient survival using decimals does not make sense. Regrettably, this article does not go give relevant information on previous retrospective published surveys on this issue.
Author Response
Thank you very much for reviewing the revised manuscript. We updated the referenced statistic from CBTRUS from 7.2% to 7% to address the concern regarding decimal point.